# Flavonoid Metabolism in *Tetrastigma hemsleyanum* Diels et Gilg Based on Metabolome Analysis and Transcriptome Sequencing

**DOI:** 10.3390/molecules28010083

**Published:** 2022-12-22

**Authors:** Yan Bai, Lingtai Jiang, Zhe Li, Shouzan Liu, Xiaotian Hu, Fei Gao

**Affiliations:** 1Zhejiang Provincial Key Laboratory of Resources Protection and Innovation of Traditional Chinese Medicine, Hangzhou 311300, China; 2College of Food and Health, Department of Traditional Chinese medicine, Zhejiang Agriculture & Forestry University, Hangzhou 311300, China; 3State Key Laboratory of Subtropical Silviculture, Zhejiang Agriculture and Forestry University, Hangzhou 311300, China; 4Botanical Garden, Zhejiang Agricultural and Forestry University, Hangzhou 311300, China

**Keywords:** *Tetrastigma hemsleyanum* Diels et Gilg, flavonoid biosynthesis, key genes, photosynthesis-antenna proteins, chalcone isomerase, UDP-glycose flavonoid glycosyltransferase

## Abstract

*Tetrastigma hemsleyanum* Diels et Gilg, known as a “plant antibiotic”, possesses several attractive properties including anti-inflammatory, anti-tumor, and antioxidant effects, with its efficacy being attributed to flavonoids. However, the flavonoid biosynthesis of *T. hemsleyanum* has rarely been studied. In this study, we investigated the flavonoid metabolism of *T. hemsleyanum* through metabolome analysis and transcriptome sequencing. The metabolomic results showed differences in the flavonoids of the leaves and root tubers of *T. hemsleyanum*. A total of 22 flavonoids was detected, and the concentrations of most flavonoids in the leaves were higher than those in the root tubers. Transcriptome analysis revealed that differentially expressed genes (DEGs) in the leaves and root tubers were enriched in photosynthesis-antenna proteins. Pearson correlation analysis indicated that the expression levels of chalcone isomerase (CHI) and UDP-glycose flavonoid glycosyltransferase (UFGT) were highly correlated with the concentrations of most flavonoids. Further, this study found that the photosynthesis-antenna proteins essentially contributed to the difference in the flavonoids in *T. hemsleyanum*. The gene expressions and concentrations of the total flavonoids of leaves and root tubers in Hangzhou, Jinhua, Lishui, and Taizhou in Zhejiang Province, China, showed that CHI (CL6715.Contig1_All, Unigene19431_All, CL921.Contig4_All) and UFGT (CL11556.Contig3_All, CL11775.Contig1_All) were the potential key genes of accumulation of most flavonoids in *T. hemsleyanum*.

## 1. Introduction

*Tetrastigma hemsleyanum* Diels et Gilg is a rare Chinese herbal medicine that is also known as a “plant antibiotic”. It is a folk medicine mainly produced in Zhejiang Province, and is widely distributed in Hangzhou, Jinhua, Lishui, Taizhou, and Ningbo, among other regions. Traditionally, it can be used to cure fever, detoxify, and promote blood circulation [1]. *T. hemsleyanum* has gradually become widely accepted due to its anti-tumor, anti-inflammatory, and antioxidant effects. The Hua Shi Xuan Fei mixture, a prescription that makes a difference in the clinical treatment of COVID-19 (approval number of Zhejiang medicine, Z20200026000), is mainly composed of *T. hemsleyanum* [2]. Considering its use in medicine, *T. hemsleyanum* has been excessively developed for the multi-application of its root tubers over the last 3–5 years, which has led to a sharp decline in its natural resources [3,4]. Further, its chaotic quality management has resulted in the medicine being characterized by inconsistent curative effects. Consequently, we have focused our efforts toward flavonoid biosynthesis research in order to improve the quality of *T. hemsleyanum.*

Analysis of *T. hemsleyanum*’s metabolic components shows numerous chemical components in the whole plant, which mainly include flavonoids, triterpenoids, organic acids, and fatty acids [5]. Flavonoids, a type of defense substance, are produced when *T. hemsleyanum* is subjected to environmental stresses. Flavonoids are generally considered to be the main active constituent of *T. hemsleyanum.* However, reports on *T. hemsleyanum* in recent years have mainly focused on variety optimization, cultivation technology management [6,7], active ingredient extraction, and related pharmacological experiments. Information on the flavonoid composition and biosynthesis genes of *T. hemsleyanum* is limited. Phenylalanine ammonia lyase (PAL), a key bridge connecting primary metabolism, is the rate-limiting enzyme in the first step of phenylpropanoid metabolism [8]. The biosynthesis of flavonoids is branched from the general phenylpropanoid pathway via the catalysis of chalcone synthase (CHS) and chalcone isomerase (CHI) [9]. CHS is the first rate-limiting enzyme in the synthetic pathway and belongs to the plant polyketide synthase superfamily [10]. It is mainly responsible for the binding of one molecule of 4-coumaroyl-CoA to three molecules of malonyl-CoA. CHI, which is the second rate-limiting enzyme involved in the biosynthetic pathway of flavonoids, catalyzes the intramolecular cyclization reaction and converts bicyclic chalcone into tricyclic (2S)-flavanone [11]. In addition to PAL, CHS, and CHI, many enzymes are involved in the flavonoid synthesis pathway, such as flavonol synthase (FLS), flavone synthase (FNS), and UDP-glycose flavonoid glycosyltransferase (UFGT). FLS is a key enzyme in the synthesis of flavonols, and it synthesizes target products using dihydroflavonols as substrates in *Arabidopsis thaliana* (L.) Heynh [12]. FNS is a key enzyme in the synthesis of flavones; it introduces a double bond between the C2 and C3 positions of flavanones to convert flavanones into flavones [13]. The UFGT enzyme is the final gene of the anthocyanin synthesis pathway, and it transfers the glucosyl moiety from UDP-glucose to the 3-hydroxyl group of anthocyanidins [14]. 

The co-application of transcriptome and metabolome analyses has become an effective method to investigate biosynthetic pathways; it can identify novel functional genes and key genes and classify the regulatory mechanisms of pathways [15]. In this study, we utilized the abovementioned method to explore the mechanism of flavonoid accumulation in the leaves and root tubers of *T. hemsleyanum*, as well as to explore the mechanism of *T. hemsleyanum*’s flavonoid metabolic pathway and its core genes. We also clarified the synthesis mechanism of the flavonoids in the leaves and root tubers of *T. hemsleyanum* and discovered the key genes involved in the biosynthesis of these flavonoids.

## 2. Results

### 2.1. Flavonoid Components Differ in Leaves and Root Tubers of T. hemsleyanum

A total of 22 flavonoid monomers was identified in *T. hemsleyanum*, and their accumulation was obviously different in its leaves and root tubers. The principal component analysis (PCA) score plot showed that the two organs were markedly separated, and the metabolome sequencing analysis had good repeatability. Indeed, PC1 and PC2 (82.2% and 8.1%, respectively) were clearly separated among the root tubers and leaves (Figure 1A).

The cluster heatmap of the 22 metabolites (Figure 1B) showed that the accumulation levels of various components including kaempferol, eriodictyol, astragaline, apigenin, and isorhamnetin were clustered together. In the metabolome analysis, luteolin, luteoloside, hesperidin, rutin, lonicerin, baicalin, and apigenin-7-*O*-glycoside were found to only be contained in the leaves, and kaempferol was only found to be contained in the root tubers. Concentrations of vitexin, orientin, lonicerin, rutin, isoquercitrin, luteolin, luteoloside, hesperidin, and naringin in the leaves were higher than those in the root tubers. For instance, the concentration of vitexin, which reached 418.828 μg·g^−1^, was higher in the leaves than in the root tubers. The concentration of orientin in the leaves was 3283.919 times higher than that in the root tubers. By contrast, the astragaline content in the root tubers was 27.636 times higher than that in the leaves (Figure 1C,D and Appendix A).

### 2.2. Transcriptome Sequencing of Leaves and Root Tubers of T. hemsleyanum

A total of 47,471,677 bp non-repetitive sequences was detected in the leaves, and 53,338,272 bp were found in the root tubers. The average lengths of the transcripts were 895.03 and 875.07 bp. The longest non-repetitive gene was 15,654 bp, and the shortest was 301 bp. Among them, the length of the N50 of the leaves was 1275 bp, and the length of the N50 of the root tubers was 1236 bp. These results indicated a high-quality assembly (Table 1).

To understand their relevant biological significance, the assembled unigenes were searched against seven functional databases (NR, GO, KOG, KEGG, and SwissProt) and annotated. As shown in Figure 2, 61,820 functional genes were annotated after searching the transcriptome sequences in KOG, NR, SwissProt, KEGG, and GO. Of the 61,820 annotated genes, 28,757 were found in leaves and 33,063 were observed in the root tubers. A total of 28,757 functional genes in the leaves were jointly annotated in five functional databases (Figure 2A). In the root tubers, 33,063 functional genes were annotated in all databases (Figure 2B). 

The assembled unigenes were statistically annotated for three biological functions in GO using Blast2GO. The study showed 22 groups in biological processes, 25,003 in cellular components, and 303 in molecular function (Figure 3A). The unigenes that matched with the KEGG database were divided into seven branches, namely, cellular processes, environmental information processing, genetic information processing, human diseases, metabolism, organic systems, and drug development. In addition, 1751 genes were related to secondary metabolism, and 2517 genes were related to environmental adaptation (Figure 3B).

### 2.3. Differentially Expressed Genes of Leaves and Root Tubers of T. hemsleyanum

Through our further study of the various differentially expressed genes (DEGs) in the leaves and root tubers of *T. hemsleyanum*, a result similar to that of the differential metabolite analysis emerged. Compared with the leaves of *T. hemsleyanum*, 67,345 DEGs were identified in the root tubers of *T. hemsleyanum*, including 6876 up-regulated genes, 14,277 down-regulated genes, and 46,192 non-DEGs (Figure 4A). When they were divided into the seven KEGG pathways used for the assembled unigenes, the majority of the DEGs were obviously enriched in the metabolism-related pathways including cellular processes, environmental information processing, genetic information processing, human diseases, metabolism, organismal systems, and drug development (Figure 4B). The 20 most significant pathways in the leaves versus the roots are listed in Figure 4C. Additionally, the DEGs in the two different medicinal portions were enriched in various pathways, including the photosynthesis, photosynthesis-antenna proteins, flavone and flavonol biosynthesis, phenylpropanoid biosynthesis, anthocyanin biosynthesis, isoflavonoid biosynthesis, biosynthesis of the secondary metabolites, plant mitogen-activated protein kinase (MAPK) signaling, flavonoid biosynthesis, and plant hormone signal transduction pathways. Among the 20 most significant pathways, the biosynthesis of the secondary metabolite pathway was enriched with the highest number of DEGs, reaching 2250 DEGs. Additionally, DEG enrichment was most significant in the plant hormone signal transduction pathway. By contrast, DEG enrichment was the most insignificant in alpha-linolenic acid metabolism. The greatest enrichment of DEGs was found in the photosynthesis-antenna proteins pathway (Figure 4C). A total of 25 DEGs belonging to the chlorophyll a/b-binding protein family was enriched in the photosynthesis-antenna proteins pathway (Figure 4D). 

### 2.4. Analysis Correlation between Flavonoid Biosynthetic Gene Expression and Flavonoid Concentration

In this part, the detected flavonoid monomers were divided into six classes of flavonoids according to their basic skeletons (Figure 5). Flavones, flavone glycosides, flavonols, flavonol glycosides, flavanones, and flavanone glycosides constituted the major fractions of the flavonoids in *T. hemsleyanum*. As illustrated in Figure 5, the flavones included vitexin, orientin, luteolin, apigenin, diosmetin, and chrysoeriol. The flavone glycosides were composed of lonicerin, luteoloside, baicalin, and apigenin-7-*O*-glycoside. The flavonols were composed of myricetin, kaempferol, and isorhamnetin. The flavonol glycosides consisted of isoquercitrin, rutin, astragalin, and isohamnetin-3-*O*-glucoside. The flavonones included naringenin and eriodictyol. Finally, the flavanone glycosides consisted of naringin and hesperidin.

A rough flavonoid metabolism pathway of *T. hemsleyanum* is listed in Figure 6; it contained 11 key genes related to flavonoid synthesis and 22 flavonoid metabolites. The heatmaps of the expression of synthetases revealed that the expression levels of CHI, UFGT, and FNS in the leaves were higher than those in the roots. For instance, the log_10_ (FPKM) value of CHI1 (CL6715.Contig1_All) in the leaves was higher than that in the root tubers by 0.213 units, and the log_10_ (FPKM) value of UFGT1 (CL11556.Contig3_All) in leaves was 1.746 times higher than that in root tubers. The expression levels of C4H, 4CL, F3′H, and F3′5′H in the leaves and root tubers were similar. The expression levels of PAL, CHS, F3H, and FNS in the roots were higher than those in the leaves. For instance, the log_10_ (FPKM) value of CHS (CL1413.Contig5_All) in the root tubers was 1.554 times higher than that in the leaves.

Next, Pearson correlation analysis of the relationship between flavonoid monomer concentrations and the typical FPKM values of the key genes involved in flavonoid biosynthesis were carried out. Interestingly, 15 flavonoids, namely vitexin, orientin, lonicerin, rutin, isoquercitrin, luteolin, luteoloside, hesperidin, naringin, eriodictyol, myricetin, baicalin, chrysoeriol, diosmetin, and quercitrin, were highly correlated with the gene expression of UFGT (Figure 7). Moreover, 12 flavonoids, including apigenin-7-glycoside, quercitrin, eriodictyol, luteolin, isoquercitrin, hesperidin, chrysoeriol, and naringin, were highly correlated with the gene expression of CHI. Three flavonoids, namely, astragaline, naringenin, and kaempferol, were highly correlated with the gene expression levels of CHI, PAL, and C4H. Orientin, isorhamnetin-3-*O*-glucoside, myricetin, baicalin, and diosmetin were moderately correlated with CHI. Astragaline, naringenin, and kaempferol were moderately correlated with F3H, F3′H, and FLS. Isorhamnetin-3-*O*-glucoside, apigenin, and apigenin-7-*O*-glycoside were moderately correlated with UFGT. A low correlation between the concentrations of astragaline, naringenin, and kaempferol and the gene expression of FNS was also found.

### 2.5. Validation of the DEGs

Next, we used quantitative real-time PCR (qRT-PCR) to determine the expression of 12 DEGs in the root tubers and leaves of *T. hemsleyanum* (Figure 8A). Consistent with the transcriptome analysis results, the flavonoid-synthesis-related key genes consisted of three CHIs (CL6715.Contig1_All, Unigene19431_All, and CL921.Contig4_All) and two UFGTs (CL11556.Contig3_All and CL11775.Contig1_All) whose gene expression in the leaves were much higher than those in the root tubers. The gene expression of four CHSs (CL1413.Contig5_All, CL1413.Contig12_All, CL1413.Contig4_All, and CL1413.Contig11_All), one FNS (CL21915.Contig3_All) and two PALs (CL2379.Contig9_All and CL2379.Contig4_All) in the root tubers was much higher than that in the leaves, a finding which concurred with the RNA-seq analysis results. In Figure 8B, the scatter plot of the correlation between RNA-seq and relative gene expression indicated a positive correlation (R^2^ = 0.8171). This result showed that the RNA-seq data were reliable.

### 2.6. Key Gene Expression and Total Flavonoid Content Trends for T. hemsleyanum from Different Locations

*T. hemsleyaum* specimens from Hangzhou, Jinhua, Lishui, and Taizhou were collected to detect their concentration of total flavonoids and expression levels of key genes including CHI 1 (CL6715.Contig1_All), CHI 2 (Unigene19431_All), CHI 3 (CL921.Contig4_All), UFGT 1 (CL11556.Contig3_All), and UFGT 2 (CL11775.Contig1_All). The flavonoid synthesis key gene expression results found that these key genes’ relative expressions in the leaves and root tubers showed similar trends to those obtained from the RNA-seq data. Furthermore, the maximum difference was observed in the expression of CHI 2 in the leaves and root tubers from Hangzhou, Jinhua, and Taizhou (Figure 9A). As shown in Figure 9B, across the four locations, the concentrations of the total flavonoids in the leaves were significantly higher than those in the root tubers. Further, a comparison of the total flavonoid content of the specimens from the four locations showed that the leaves and root tubers of Lishui *T. hemsleyanum* were of a better quality than the others, followed by Taizhou.

## 3. Discussion

A unified standard for the medicinal part of *T. hemsleyanum* has been lacking for several years in China. *Flora of China* records that the whole plant can be used for medicine, whereas Zhejiang Province, as the main production area of *T. hemsleyanum*, issued *Standards of Processing Chinese Crude Drugs*, which classifies the root tubers as drug-parts [22,23]. In the present study, metabolomic analysis of the whole plant indicated that the species and concentration of flavonoids in *T. hemsleyanum’s* leaves were higher than those in the root tubers. Interestingly, a great deal of anti-inflammatory and anti-tumor flavonoid metabolites were found in the leaves. Thus, we suggest that *T. hemsleyanum*’s leaves should also be used for medicine. Transcriptome analysis revealed that the DEGs in the photosynthesis-antenna proteins played a role in flavonoid metabolism in the leaves and root tubers. To date, few studies have thoroughly analyzed the flavonoid biosynthesis of *T. hemsleyanum*. In the present study, we explored flavonoid metabolism, finding key genes called CHI and UFGT via metabolomic analysis and transcriptome sequencing.

### 3.1. Key Role of DEGs in Photosynthesis-Antenna Proteins in Flavonoid Metabolism in Leaves and Root Tubers

In this study, metabolomic analysis showed that the concentrations of most flavonoids in the leaves were higher than those in the root tubers. Compared with the flavonoids in the root tubers, flavones and flavone glycosides were mostly produced in the leaves. Zhang et al. also found identical occurrences [24]. Further, many anti-inflammatory and anti-tumor ingredients were detected in the leaves, such as luteolin, luteoloside, hesperidin, baicalin, chrysoeriol, diosmetin, apigenin, vitexin, orientin, lonicerin, myricetin, eriodictyol, quercitrin, and isoquercitrin. Analysis of metabolites also indicated that the major flavonoids in the leaves of *T. hemsleyanum* were vitexin and orientin.

KEGG enrichment analysis indicated that the flavonoids in the leaves and root tubers were enriched in the photosynthesis, photosynthesis-antenna proteins, plant MAPK signaling, plant hormone signal transduction, phenylpropanoid biosynthesis, biosynthesis of the secondary metabolite, flavone and flavonol biosynthesis, and flavonoid biosynthesis pathways. The DEGs in these pathways may jointly play a role in regulating key enzymes and may ultimately lead to differences in the concentrations of flavonoid metabolites between the leaves and root tubers. Photosynthesis and photosynthesis-antenna proteins were found to be the key pathways involved in flavonoid synthesis. About 95% of plant biomass accumulation was attributed to photosynthesis, and the regulation of photosynthesis-antenna proteins was found to regulate the ability of the leaves to capture solar energy and transfer energy to reaction centers [25]. Located on the thylakoid membrane of chloroplasts, photosynthesis-antenna proteins were found to have a high expression in the leaves and low expression in the root tubers [26]. Different expression of DEGs enriched in the photosynthesis-antenna protein pathway between the root tubers and leaves resulted in varying levels of light energy being captured, which affected the synthesis of the C skeleton [27]. The C skeleton plays a key role in flavonoid synthesis. Wang et al. [28] found that the down-regulation of photosynthesis-antenna protein-related genes in citrus infected with *Candidatus Liberibacter asiaticus* resulted in a significant decrease in flavonoids and phenylalanine derivatives. Additionally, the leaves and root tubers of *T. hemsleyanum* are exposed to ultraviolet (UV) radiation with distinct intensities, also affecting photosynthesis-antenna proteins’ roles in energy capture and flavonoid biosynthesis. Hong et al. indicated that chlorophyll a/b-binding proteins are initially downregulated after shading, which is consistent with the expression pattern of genes including CHI and 3GT in the anthocyanin synthesis pathway [29]. Our preliminary study found that appropriate doses of UV-B and UV-C radiation (30 min–3 h) induced active stress and contributed to the accumulation of flavonoids [30]. Furthermore, we previously conducted different shading experiments on *T. hemsleyanum* and found that light intensity affected its flavonoid accumulation [15]. In the two different medicinal portions studied in this work, the DEGs in the photosynthesis-antenna proteins pathway affected the expression of DEGs in the secondary metabolite biosynthesis, flavonoid biosynthesis, and flavone and flavonol biosynthesis pathways. This phenomenon led to the discrepancy in flavonoids observed in the two studied medicinal portions. 

According to the literature, MAPKs have important functions in abiotic stresses including salt stress, drought stress, and UV light [31,32,33]. MAPKs are also involved in the response to plant hormones [34]. Plant hormones regulate flavonoid metabolism by directly regulating the expression of key genes related to flavonoid biosynthesis [35,36]. However, whether MAPKs and phytohormones are involved in flavonoid biosynthesis in root tubers and leaves of *T. hemsleyanum* requires further study.

### 3.2. CHI and UFGT Are Critical Genes for Flavonoid Biosynthesis in T. hemsleyanum

Flavonoid metabolites may play a critical role in plant growth, as well as plant resistance to biological and abiotic stresses, such as extreme temperature, drought, UV light, nutrient deficiencies, pathogens, and herbivores; however, their biosynthetic pathways and core genes in *T. hemsleyanum* are yet to be uncovered [37]. 

Following Peng’s research [19], we plotted the framework of the flavonoid metabolic pathway of *T. hemsleyanum* and classified its flavonoid metabolites. We concretized the metabolic pathway reference to what is known of flavonoid metabolic pathways in plants such as citrus, celery, and buckwheat. PAL, C4H, and 4CL are jointly involved in the synthesis of P-coumaroyl-coA. CHS contributes to the production of chalcone. Chalcone can transform into naringenin by catalyzing the cyclization reaction with CHI. Naringenin is glucosylated and rhamnosylated by 7-glucosyltransferase and 1-2-rhamnosyltransferase to naringin [16], and hesperidin is also derived from naringenin. Naringenin’s B-3′ position is hydroxylated by F3′H to accumulate eriodictyol. F3′5′H can use eriodictyol to generate flavanones. FNS catalyzes the conversion of flavanones to various flavones including vitexin, orientin, apigenin, and chrysoeriol. The production of lonicerin, baicalin, and luteoloside undergoes glycosylation. Apigenin is methylated and glycosylated to diosmetin and apigenin-7-*O*-glycosides by OMT and UFGT, respectively [21]. By catalyzing with F3′H or F3′5′H, apigenin transforms into luteolin [21]. In another synthesis branch, F3H catalyzes flavanones at the C-3 position to accumulate dihydroflavonol. The conversion of dihydroflavonol to anthocyanidins with a stable structure is catalyzed by DFR, LDOX, and UFGT. FLS can also use dihydroflavonol as a substrate to produce various flavonols including myricetin, kaempferol, and quercitrin. Quercitrin is methylated and glycosylated to isoquercitrin and isorhamnetin, respectively [17]. By catalyzing with 1,6-Rhamnosyltransferase, isorhamnetin transforms into rutin [18]. Subsequently, flavonol is glycosylated to flavonol glycosides such as isorhamnetin-3-*O*-glucoside and astragalin.

In our study, some key genes related to the accumulation of flavonoids were found using Pearson correlation analysis. The total flavonoid and highest flavonoid concentration trends were similar to those of the gene expressions of CHI (CL6715.Contig1_All, Unigene19431_All, and CL921.Contig4_All) and UFGT (CL11556.Contig3_All and CL11775.Contig1_All) in *T. hemsleyanum*. Regarding the key gene expressions and total flavonoid content of *T. hemsleyanum* from different locations, the results indicated that CHI and UFGT were the candidate genes for the flavonoid biosynthesis pathway. Nam Il Park et al. also demonstrated that transgenic hairy roots with high SbCHI expression mediated by *Agrobacterium* rhizogenes can synthesize many flavones [38]. Although UFGT is a downstream enzyme that is generally considered a key gene related to the synthesis of anthocyanin, some studies have indicated that UFGT also plays an important role in the synthesis of other flavonoids [39,40]. The catalytic rate of the enzyme and the consumption rate of the substrate may lead to this phenomenon.

## 4. Materials and Methods

### 4.1. Plant Materials

Fresh and healthy three-year-old *T. hemsleyanum* plotted plants from Hangzhou, Jinhua, Lishui, and Taizhou were collected in summer from Zhejiang Agriculture and Forestry University (Hangzhou Zhejiang, 30°15′32″ N, 119°43′23.14″ E) in summer. Sample collection was repeated three times. The collected root tubers and leaves were immediately frozen in liquid nitrogen and then stored at −80 °C.

### 4.2. Determination of Flavonoid Metabolomes

Approximately 0.1 g of root tubers or leaf powder was placed accurately in a 2.0 mL EP tube, and to this was added 1.5 mL of 80% methanol (*v*/*v*). Vitexin, orientin, afzelin, rutin, isoquercitrin, luteolin, luteoloside, isoorientin, naringin, isorhamnetin-3-*O*-glucoside, isorhamnetin, astragaline, eriodictyol, taxifolin, isovitexin, calycosin-7-*O*-glucoside, hispidulin, naringenin, apigenin, quercitrin, and apigenin-7-*O*-glycoside were used as reference substances, and their fabrication processes were consistent with the abovementioned process. The samples were sonicated for 30 min, and the extractions were filtered through a 0.22 μm PTFE membrane to obtain the filtrate. The obtained filtrate was sent to Kyushu Kexin (Beijing, China) Testing Technology Co., Ltd. for metabolome determination. The analytic conditions were as follows:

Chromatographic conditions: The testing instrument was an API 6500 quadrupole-linear ion trap mass spectrometer (Framingham, MA, USA). An ACQUITYUPLCBEH C18 column (2.1 mm × 100 mm, 1.8 μm) was used with 0.1% formic acid-water (A)-0.1% formic acid-acetonitrile (B) as moving phase, gradient elution (0 min, 5% B; 1 min, 25% B; 3.5 min, 40% B; 4.5 min, 60% B), and flow rate of 0.6 mL/min. The column temperature was 40 °C, and the injection volume was 1 μL.

Ion source: Turbo V; ionization mode: ESI-; gas curtain gas (CUR) volume flow rate: 30 L/min; spray voltage (IS): −4500 V; nebulizing gas (GS1) volume flow rate: 55 L/min; heating assist gas (GS2) volume flow rate: 55 L·min^−1^.

Acquisition method: multiple reaction monitoring mode (MRM), ionization temperature (TEMP): 550 °C.

Investigation of linear relationship: The peak area was measured according to the above conditions, and the regression equation was obtained using the injection concentration X (μg/mL) as the horizontal coordinate and the chromatographic peak area Y as the vertical coordinate as shown in Table 2.

### 4.3. RNA Extraction and Transcriptome Sequencing

Total RNA was extracted using the plant total RNA extraction kit (Vazyme). We rapidly ground 50 mg of fresh leaves and root tubers in liquid nitrogen and then added buffer PSL for lysis. The RNA of *T. hemsleyanum* was extracted according to the manufacturer’s instruction. RNA purity, concentration, and integrity were tested with BERTHOLD Colibri LB 915 Microvolume (Berthold, Germany) and Bio-Rad PowerPac (Hercules, CA, USA). In addition, 7μL of RNA was used for reverse transcription into cDNA reference in accordance with the TaKaRa reverse transcription kit. The extracted cDNA samples of *T. hemsleyanum* of Lishui were sent to Sangon Biotech (Xuhui, SHH, China) Co., Ltd., for transcriptome sequencing. Libaries were constructed using an Agilent 2100 Bioanalyzer and ABI StepOnePlus Real-Time PCR System for quality control. The qualified libraries were sequenced on the Illumina HiSeq platform. After the raw data of low quality, adapter contamination, and high unknown N concentration were removed, clean reads were obtained. 

### 4.4. De Novo Assembly and Sequence Clustering

Trinity v2.5.1 was used for de novo assembly, and Tgicl was used for clustering the assembled transcripts to obtain unigenes. The clean reads were segmented into a large number of individual de Bruijn plots from which full-length transcript shear isoforms were extracted. The transcripts were clustered and redundantly processed with Tgicl. For multiple samples, Tgicl was used again for each unigene of each sample to obtain the final unigene for subsequent analysis.

### 4.5. FPKM Calculation and DEGs Screening

The FPKM value calculation formula is as follows: (1)$$\mathrm{FPKM}=\frac{1,000,000\times C}{N\times L}\times 1000$$
C is the number of fragments compared with the transcript, N is the total number of reads of all transcripts, and L is the number of transcript bases.

Differential expression of genes was detected using Audic’s [41] sequencing-based method. In this research, false discovery rate (FDR) ≤0.001 and fold change ≥2 were used as criteria for the determination of DEGs.

### 4.6. Functional Annotation and Classification of Genes

NR, GO, KOG, KEGG, and SwissProt belong to functional databases. In this section, the functional annotation and classification of unigenes were performed by Blast v2.2.23. 

Differentially expressed genes were annotated and classified into the KEGG pathways. According to the annotation and classification results, DEGs were subjected to biological pathway classification. In the meantime, phyper in the R package was used for enrichment analysis. FDR ≤0.01 was used as a criterion for the rectification of *p*-values.

### 4.7. Quantitative Real-Time PCR

The primer sequences are listed in Table 3. These primers were designed using an online site (https://www.genscript.com/tools/real-time-pcr-taqman-primer-design-tool?act=register_success, accessed on 7 June 2022) according to the CDS sequences and synthesized by SUNYA. MDH was used as a reference gene [42]. A quantitative real-time PCR experiment using specialized reagents (TaKaRa) followed the CFX96 Real-Time PCR Detection System instructions, and three technical replicates were prepared for each sample. The 25 μL system contained 2 μL of cDNA, 12.5 μL of TB Green *Premix Ex Taq* Ⅱ(Tli RNaseH Plus), 8.5 μL of sterile water, and 2 μL of forward and reverse primers. The two-step PCR reaction was incubated at 95 °C for 30 s for 40 cycles and then incubated at 95 °C for 5 s and 60 °C for 30 s. Gene expression was calculated by 2^−ΔΔCt^, and special amplification of primers was judged with melting curve analysis. 

### 4.8. Total Flavonoid Determination

After being freeze-dried, the root tubers and leaves of *T. hemsleyanum* were ground into a powder that could be passed through an 80 mesh sieve net. The total flavonoid content was determined by UV spectrophotometry according to a method that improved on that of Han et al. [6].

Linear relationship investigation: With rutin as the standard, the test scheme was consistent with the above procedure. The absorbance was measured at 501 nm (the maximum absorption wavelength obtained by scanning the field). With absorbance A as the ordinate and reference mass concentration C as the abscissa, the linear regression equation was A = 12.045 C + 0.0019, r^2^ = 0.99997.

### 4.9. Statistical Analysis

Metabolome and q-PCR results were calculated using Excel 2020 (Redmond, WA, USA), and data were expressed as mean ± standard error (SE). One-way analysis of variance (ANOVA) and *t*-test with IBM SPSS statistical software 24.0 (Armonk, NY, USA) were used for statistical significance analysis. As for RNA-seq results, the R package was used for analysis and drawing. PCR analysis, heatmaps, bar graphs, scatter plots, and Pearson correlation analysis were performed using Origin 2021 (Northampton, MA, USA). 

## 5. Conclusions

Metabolite analysis and transcriptome sequencing were used to explore the mechanism of flavonoid accumulation in the leaves and roots and the flavonoid metabolic pathways of *T. hemsleyanum*, as well as to explore the core genes of flavonoid biosynthesis. Given that the active ingredient levels of flavonoids in the leaves were higher than those in the root tubers, we suggest that the leaves should be given more attention in future research on *T. hemsleyanum*. We found regulatory proteins called photosynthesis-antenna proteins that play a key role in the flavonoid biosynthesis pathway of *T. hemsleyanum*, which researchers have previously ignored. This study also illustrated that CHI (CL6715.Contig1_All, Unigene19431_All, and CL921.Contig4_All) and UFGT (CL11556.Contig3_All and CL11775.Contig1_All) were critical genes in most of the flavonoid metabolism pathways of *T. hemsleyanum*. This study provides a theoretical basis for the development and utilization of the leaves and the improvement of the flavonoids of *T. hemsleyanum*. 

## Figures and Tables

**Figure 1 molecules-28-00083-f001:**
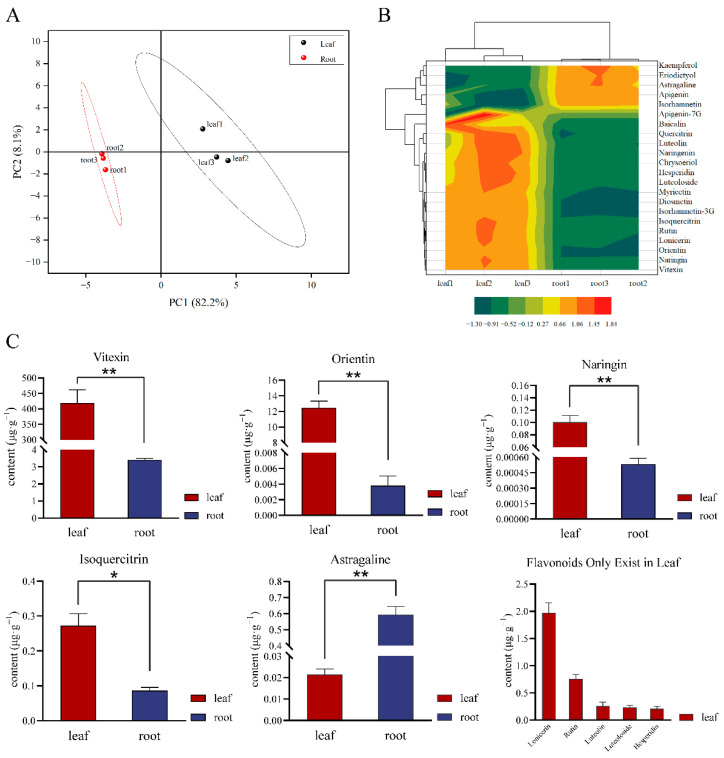
Metabolomic sequencing analysis of flavonoids. (**A**) PCA score plot of leaves and root tubers of *T. hemsleyanum*; (**B**) Cluster contour heatmap of 22 metabolites; (**C**) Concentration of flavonoid active ingredients in different parts of *T. hemsleyanum*; (**D**) Flavonoids only existing in leaves. (Concentration of flavonoids in leaves or root tubers more than 0.1 μg·g^−1^; different symbol means, * represents *p* ≤ 0.05, ** represents *p* ≤ 0.01.)

**Figure 2 molecules-28-00083-f002:**
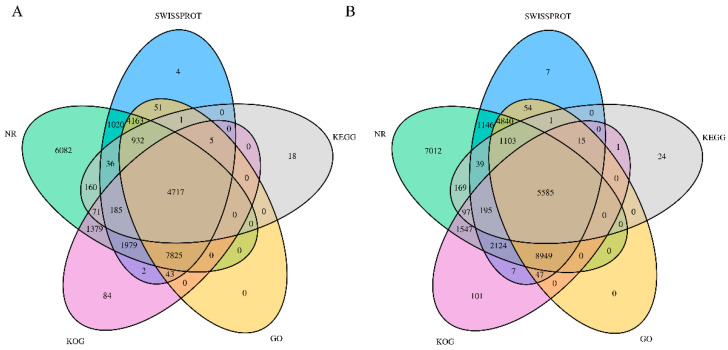
Venn diagrams of unigene annotations. (**A**) Venn diagram of KOG, NR, SwissProt, KEGG, and GO gene database annotation from leaves of *T. hemsleyanum*; (**B**) Venn diagram of NT, NR, SwissProt, KEGG, and GO gene database annotation from root tubers.

**Figure 3 molecules-28-00083-f003:**
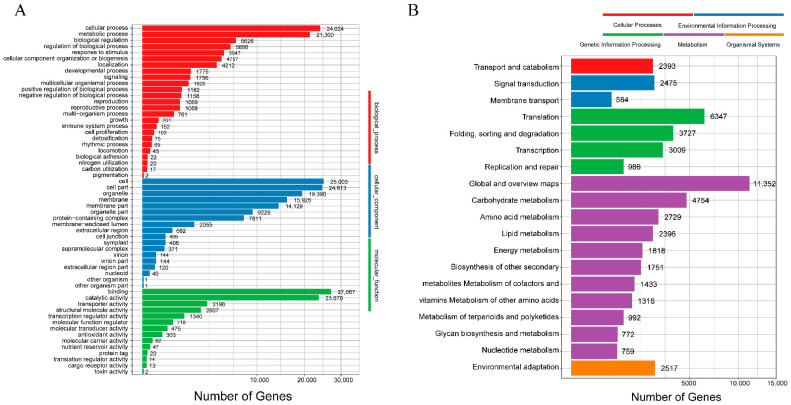
Classification of the assembled unigenes. (**A**) GO classification of unigenes; (**B**) KEGG classification of unigenes.

**Figure 4 molecules-28-00083-f004:**
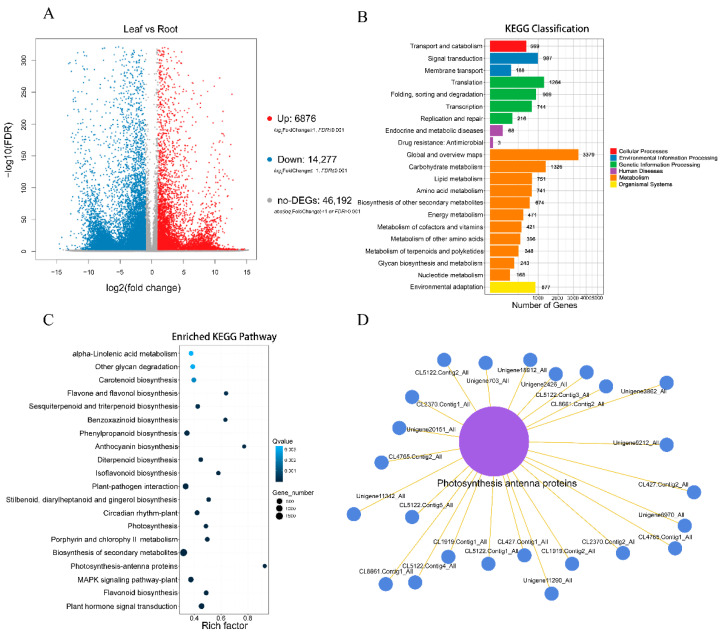
Results of DEGs in different parts of *T. hemsleyanum*. (**A**) Volcano plots for DEGs from leaves vs. roots; (**B**) KEGG classification of DEGs from leaves vs. roots; (**C**) Scatter plot of differentially expressed *T. hemsleyanum* genes in the 20 most enriched KEGG pathways (the y-axis shows KEGG pathways, and the x-axis shows rich factor; the size of a point represents the number of DEGs, and the color of a point represents its q-value); (**D**) Network of differentially expressed *T. hemsleyanum* genes enriched in the photosynthesis-antenna proteins.

**Figure 5 molecules-28-00083-f005:**
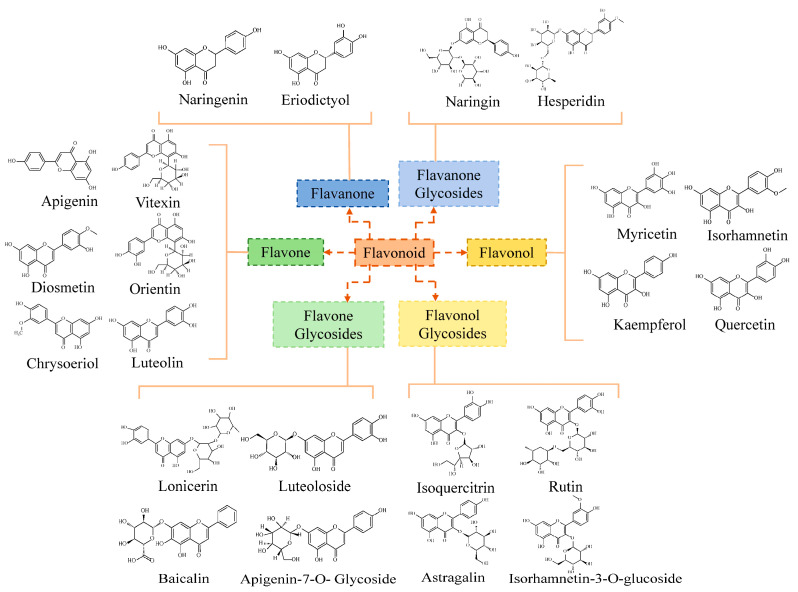
Classification map of flavonoids.

**Figure 6 molecules-28-00083-f006:**
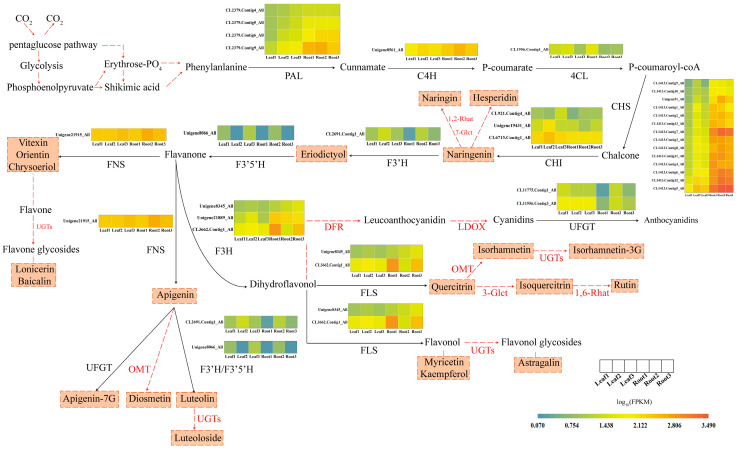
Biosynthetic pathway of flavonoids [16,17,18,19,20,21]. This pathway contains the major flavonoid metabolites and the expression of synthetases. The ingredients in the dotted box are the flavonoids found in the roots and leaves; the enzyme near the red dashed arrow was not involved in our experiment. PAL, phenylalanine ammonia lyase; C4H, Cinnamate 4-hydroxylase; 4CL, 4-coumarate-coenzyme A; CHI, chalcone isomerase; CHS, chalcone synthase; F3H, Flavanone-3-hydroxylase; F3′H, Flavanone-3′-hydroxylase; F3′5′H, Flavonoid-3′,5′-hydroxylase; FLS, Flavonol synthase; FNS, Flavone synthase; DFR, Dihydroflavonol 4-reductase; LDOX, Leucoanthocyanidin dioxygenase; 3-Glct, Flavonoid-3-*O*-glucosyltransferase; 7-Glct, Flavonoid-3-*O*-glucosyltransferase; 1,2-Rhat, 1,2-rhamnosyltransferase; 1,6-Rhat, 1,6-rhamnosyltransferase; UGTs, UDP-Glycosyltransferases; OMT, O-methyltransferase; UFGT, UDP-glycose flavonoid glycosyltransferase. The heatmaps represent log_10_ (FPKM) values of these enzymes in leaves and root tubers.

**Figure 7 molecules-28-00083-f007:**
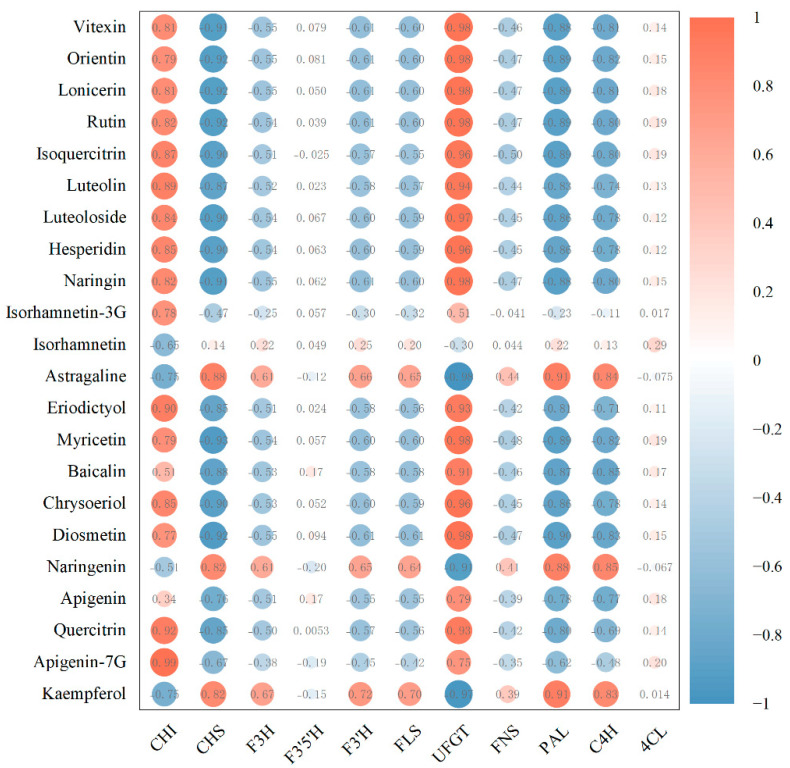
Correlations between flavonoid monomers and gene expression of related enzymes.

**Figure 8 molecules-28-00083-f008:**
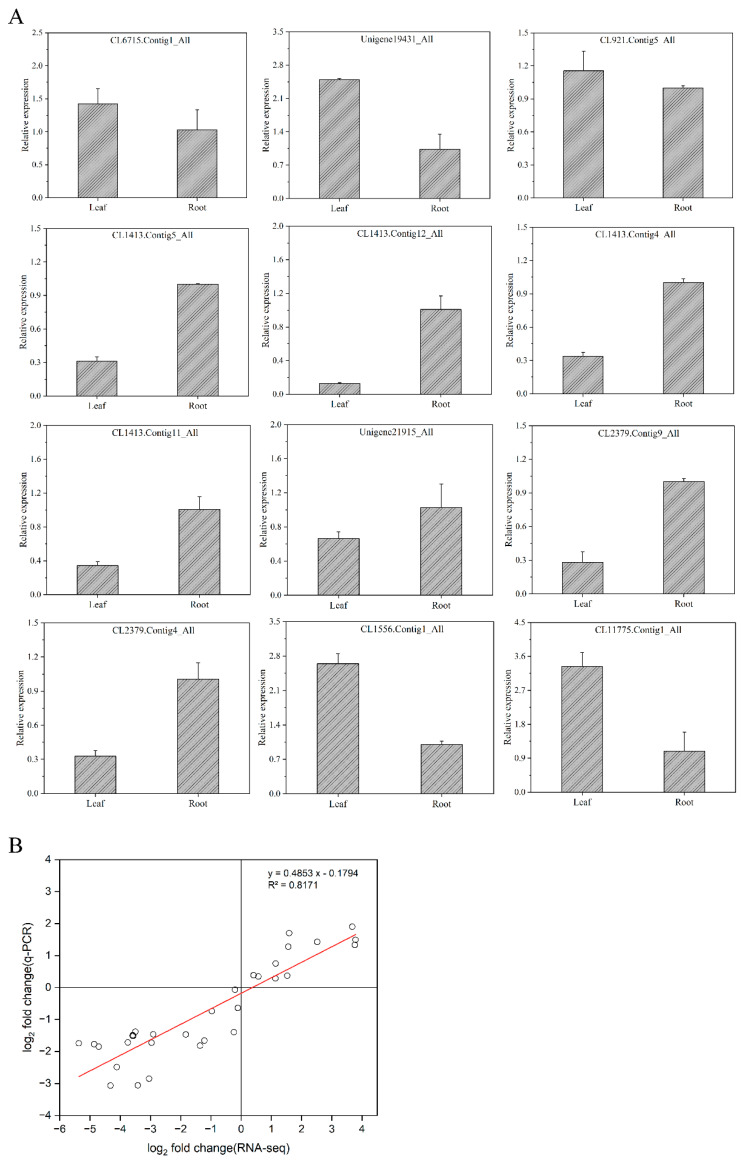
Validation of the differentially expressed genes. (**A**) Relative expressions of flavonoid-synthesis-related genes; (**B**) Correlation between RNA-seq and relative gene expression (the x-axis represents the log_2_ fold change value of the RNA-seq results, and the y-axis represents the log_2_ fold change value of the relative expression of related genes; all data were collected from three biological replicates and three technical replicates).

**Figure 9 molecules-28-00083-f009:**
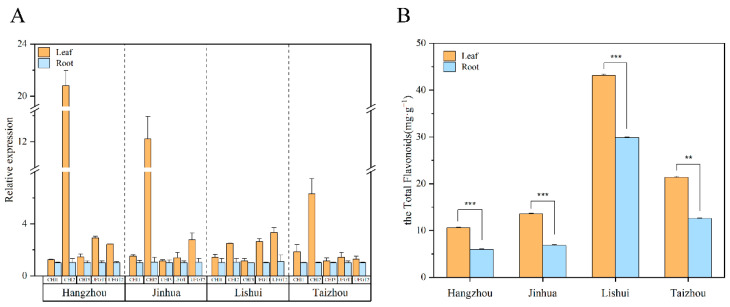
Expressions of flavonoid synthesis key genes and total flavonoid content of Zhejiang *T. hemsleyanum* from different locations. (**A**) Expression of flavonoid synthesis key genes of Zhejiang *T. hemsleyanum* from different locations (CHI 1, CL6715.Contig1_All; CHI 2, Unigene19431_All; CHI 3, CL921.Contig4_All; UFGT 1, CL11556.Contig3_All; UFGT 2, CL11775.Contig1_All.). (**B**) The concentration of total flavonoids of Zhejiang *T. hemsleyanum* from different locations (different symbols mean a significant difference, ** represents *p* ≤ 0.01, *** represents *p* ≤ 0.001).

**Table 1 molecules-28-00083-t001:** Length distribution of assembled unigenes.

Term	All > 300 bp	≥500 bp	≥1000 bp	N50	Total Length	Max Length	Min Length	Average Length
Y-Unigene	53,039	29,973	14,690	1275	47,471,677	15,654	301	895.03
G-Unigene	60,953	33,607	16,023	1236	53,338,272	15,654	301	875.07

**Table 2 molecules-28-00083-t002:** Linear relationship investigation results.

Compound	Regression Equation	r	Linear Range
Vitexin	Y = 8,542,235.9 X + 18,454,664.6	0.9987	20.000–2000.000
Orientin	Y = 9,071,337.1 X + 33,093.6	0.9996	1.000–100.000
Afzelin	Y = 9,658,807.6 X + 92,259.2	0.9993	0.001–0.050
Rutin	Y = 4,950,860.8 X + 231,551.2	0.9994	0.010–0.500
Isoquercitrin	Y = 3,356,305.1 X + 593,847.8	0.9989	0.020–1.000
Luteolin	Y = 9,109,434.3 X + 35,359.9	0.9998	0.020–1.000
Luteoloside	Y = 3,168,802.6 X + 166,068.3	0.9994	0.100–5.000
Isoorientin	Y = 4,845,698.3 X + 4,222,807.8	0.9987	1.000–100.000
Naringin	Y = 3,972,039.5 X + 64,183.6	0.9976	0.010–0.500
Isorhamnetin-3-*O*-Glucoside	Y = 8,988,517.2 X + 5,992,231.1	0.9982	0.020–1.000
Isorhamnetin	Y = 1,476,424.2 X + 2,111,521.6	0.9986	0.001–0.050
Astragaline	Y = 2,210,508.5 X + 2,507,321.4	0.9986	0.010–0.500
Eriodictyol	Y = 691,932.1 X + 84659.5	0.9991	0.010–0.500
Taxifolin	Y = 3,575,802.9 X + 284,658.7	0.9992	0.001–0.050
Isovitexin	Y = 8,542,235.9 X + 18,454,664.6	0.9987	20.000–2000.000
Calycosin-7-*O*-glucoside	Y = 22,042.7 X + 2463.8	0.9986	0.020–1.000
Hispidulin	Y = 207,676.0 X + 105,248.9	0.9993	0.020–1.000
Naringenin	Y = 12,452,147.9 X +74,631.9	0.9992	0.001–0.050
Apigenin	Y = 2,885,631.5 X + 238,287.7	0.9989	0.001–0.050
Quercitrin	Y = 12,206,204.2 X + 1,425,428.8	0.9992	0.001–0.050
Apigenin-7-*O*-Glycoside	Y = 1,806,515.0 X + 16,658.0	0.9996	0.010–0.500

**Table 3 molecules-28-00083-t003:** Primer sequences of flavonoid-related genes in different parts of *T. hemsleyanum*.

Gene	Gene ID	Tm (°C)	Primer F (5′-3′)	Primer R (5′-3′)
MDH	\	60	TGTTGCTACGACTGATGT	CCTGAGACTTGTAGATGGAA
PAL	CL2379.Contig9_All	56.02	ACCAACCATGTCCAAAGTGC	CGCCACAAGGTATGTGGAAG
	CL2379.Contig4_All	56.06	TGGTGACCTCACCTTCTCAC	GAAGCCCATCCATTCCGAAC
CHS	CL1413.Contig5_All	55.9	GGAACTGTCCTTCGAACTGC	GTCCGCGGAATGTAACAACA
	CL1413.Contig12_All	55.99	TAGCACGTTGAGCGTTTCTG	AGAGTTGGTGGCATAAGGCT
	CL1413.Contig4_All	55.71	CCAACTTGTCTCAGCAGCTC	AATCAAAGTGGGCACAGTGG
	CL1413.Contig11_All	55.89	TGTACCATCAAGGGTGCCAT	GGGCTTGGCCAACTAAAGAG
CHI	CL6715.Contig1_All	56.01	GTGCAGGGTGTGAAGTTTGT	TTTGGCTTCCTTCCAACAGC
	Unigene19431_All	56.47	ACGCCATGGATAGAGAGCAA	CCCATGGTTGAGGATTCGGA
	CL921.Contig4_All	55.96	ACTGAACACCATCCACGACT	GACCAACGAATGCCTCGAAA
FNS	Unigene21915_All	55.99	AAAGACGAACTCTCCACCGT	GATGTTCGCAACTCCGTTCA
UFGT	CL11556.Contig3_All	56.05	TTGACTTGCCTGAGTGTCCT	AACTCGGATGCTGAGTTGGA
	CL11775.Contig1_All	55.90	CAACGGCGGAATGAGCTAAA	TCTTGTGGTCCCTTCGTCAA

## Data Availability

All sequencing data are available for download at the NCBI Sequence Read Archive (SRA) under the accession number PRJNA826914.

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
