# Peer review of "Flavonoid Metabolism in Tetrastigma hemsleyanum Diels et Gilg Based on Metabolome Analysis and Transcriptome Sequencing"

_molecules, 2022, doi:10.3390/molecules28010083_

Round 1

Reviewer 1 Report (New Reviewer)

In the present study the authors have applied metabolite analysis and transcriptome sequencing to explore the mechanism of flavonoid accumulation and flavonoid metabolic pathway in Tetrastigma hemsleyanum, and the core genes of flavonoid biosynthesis. Interestingly, the analysis indicated that the concentration of flavonoids in leaves were higher than those in root tubers and a great deal of anti-inflammatory and anti-tumor flavonoid metabolites exist in leaves.

The manuscript has high novelty, significance, and scientific quality. The subject of this paper should be of interest for the readers of the Journal, the manuscript is acceptable for publication.

How many plant sample were collected and analyzed. The PCA plot depicts only 3 plants. Why?

Minor comments:

 Figures: The font size is too small, the text is hardly readable.

 Line 45. “we has focused” correctly: we have focused

 Line 103. “different symbol means a significant difference at 0.05 level.” Please clarify.

 Line 175. “Flvanonls” is a typo: Flavonols

 Materials, methods. Please specify the used HPLC-MS instrument.

 Line 456. “PCR analysis”  Do you mean PCA analysis?

Author Response

Reviewer 2 Report (Previous Reviewer 1)

In this work, entitle “Flavonoid Metabolism in Tetrastigma hemsleyanum Diels et 

Gilg based on Metabolome Analysis and Transcriptome Sequencing”, the authors report the identification of genes and metabolites of the flavonoid biosynthetic pathway in root tubers and leaves of a Chinese medicinal plant.

The manuscript needs to be improved to be accepted.

FNS is not flavonoid synthase, it is flavone synthase. Correct it along the manuscript.

In fig 2 A, B, C and D is impossible to read the numbers and words in the figures

Figure 6: include a figure A with a higher letter size. Is impossible to read the axes legends

Line 349: “The results of key gene expression and the total flavonoid content of T. hemsleyanum from different origins indicated that CHI and UFGT were the candidate genes for the flavonoid biosynthesis pathway”. Here as in several parts of the manuscript, the phrase is not precise. The manuscript needs to be fully revised by an English native to improve the grammar.  

Round 2

Reviewer 2 Report (Previous Reviewer 1)

The authors answered and corrected all the suggestions and comments, and in consequence, the manuscript has substantially improved its quality compared to the last version.  

Author Response

We appreciate the reviewer's comments. Further, we have revised the incorrect formatting and spelling in the manuscript. Here are some of the modifications:(1) We also clarified the synthesis mechanism of the flavonoids of the leaves and root tubers of T. hemsleyanum and discovered the key genes involved in the biosynthesis of these flavonoids. (2) The flavonols were composed of myricetin, kaempferol, and isorhamnetin. (3) Next, Pearson correlations analysis of the relationship between flavonoid monomer concentration and the typical FPKM values of the key genes involved in flavonoid biosynthesis were carried out. (4) A low correlation between the concentrations of astragaline, naringenin, and kaempferol and the gene expression of FNS was also found. (5) Unified standard for the medicinal part of T. hemsleyanum has been lacking for several years in China. (6) However, their biosynthetic pathways and core gene in T. hemsleyanum are yet to be uncovered.

This manuscript is a resubmission of an earlier submission. The following is a list of the peer review reports and author responses from that submission.

Round 1

Reviewer 1 Report

In this work, entitle “Flavonoid Metabolism in Tetrastigma hemsleyanum Diels et 

Gilg based on Metabolome Analysis and Transcriptome Sequencing”, the authors report the identification of genes and metabolites of the flavonoid biosynthetic pathway in root tubers and leaves of a Chinese medicinal plant.

In general, the manuscript needs to be improved to be accepted.

Some comments and suggestions for the manuscript are listed below.

Introduction:

Line 62: “as flavonol synthase (FLS), flavonol synthase (FNS)”. There is a mistake with the name of the enzymes… correct it.

Results:

Line 83: “the expression levels”. The metabolites are not expressed… I suggest correcting by accumulation levels

Line 98: “Transcriptomes of T. hemsleyanum leaves and root tubers were sequenced and assembled by Sangon Biotech (Shanghai) Co. Ltd. After the raw data of low quality, adapter contamination, and high unknown N concentration were removed, we obtained clean reads. Then, these clean reads were assembled using the Trinity program”. This sentence corresponds to methodology. Moves to the materials and methods section

Figure 2: Improve the resolution of panels C and D in figure 2. Homogenize the letter size in the caption. 

Line 168: what is nucleusus?

Figure 4: Correct the name of FLS and FNS in the caption

Discussion:

Line 234: Revise the use of capital letters in this paragraph that are not correctly used.

Additionally, discussion about the importance of flavonoids in the photosynthesis-antenna proteins is very superficial and needs more deepest analysis of the literature.

Line 284: Include the species used in the cited work

Materials and methods

Line 292: The plants used come from clonal propagation or are from different genetic backgrounds?

Line 303: The authors need to indicate the quantity of tissue used to extract the RNA. After, synthesizing the cDNA indicates the amount of RNA used. This part of material and methods require to be related in more detail

Table 3: The primers were designed for qPCR? It is necessary to indicate fragment size, and efficiency, and to include the accession number of the target gene.

And the PCA analysis how was done? The ontology classification of DEGs? 

The statistical analysis specific for each assay is not included.

In general, there is lacking information on critical methodology used in this work

In abbreviations, same mistake indicated above about flavonol synthase FLS and FNS

Reviewer 2 Report

This manuscript describes the accumulation of flavonoids in Tetrastigma hemsleyanum Diels et Gilg and annotated flavonoid biosynthetic genes based on homology and expression pattern. I have two major concerns regarding this manuscript. First, the content of the manuscript is much overlapped with some recent publications such as Shi et al. PLOS ONE, 2022. Second, annotation of genes based on homology without any functional characterization is not meaningful. Thus, I do not think this manuscript is suitable for publication in Molecules.

Other comments:

  • Line 52: Please completely describe the pathways according to figure 4. For example, some of the genes like C4H, 4CL, F3H are not described in this paragraph.
  • Figure 4: Please arrange the flavonoid structures in the same orientation.